# Physically-Constrained Adversarial Attacks on Brain-Machine Interfaces

**Xiaying Wang**
Dept. Elec. Eng. Inf. Tech.
ETH Zurich
Switzerland
xiaywang@iis.ee.ethz.ch

**Octavio R. Q. Siller**
Dept. Dept. Elec. Eng. Inf. Tech.
ETH Zurich
Switzerland
siller.rodolfo@gmail.com

**Michael Hersche**
Dept. EEIT, ETH Zurich
IBM Research Europe - Zurich
Switzerland
herschmi@iis.ee.ethz.ch

**Luca Benini**
Dept. EEIT, ETH Zurich
DEI, University of Bologna
Switzerland, Italy
lbenini@iis.ee.ethz.ch

**Gagandeep Singh**
Dept. Computer Science
UIUC, Urbana 61801
VMware Research, USA
ggnds@illinois.edu

## Abstract

Deep learning (DL) has been widely employed in brain–machine interfaces (BMIs) to decode subjects' intentions based on recorded brain activities enabling direct interaction with machines. BMI systems play a crucial role in medical applications and have recently gained an increasing interest as consumer-grade products. Failures in such systems might cause medical misdiagnoses, physical harm, and financial loss. Especially with the current market boost of such devices, it is of utmost importance to analyze and understand in-depth, potential malicious attacks to develop countermeasures and avoid future damages. This work presents the first study that analyzes and models adversarial attacks based on physical domain constraints in DL-based BMIs. Specifically, we assess the robustness of EEGNet which is the current state-of-the-art network embedded in a real-world, wearable BMI. We propose new methods that incorporate domain-specific insights and constraints to design natural and imperceptible attacks and to realistically model signal propagation over the human scalp. Our results show that EEGNet is significantly vulnerable to adversarial attacks with an attack success rate of more than 50%.

## 1 Introduction

Recent work has shown that adversarial perturbations can cause state-of-the-art (SoA) deep learning models to misbehave in various domains including vision [38, 17], NLP [24, 42], speech [34, 25], and biomedicine [15, 18]. Neural networks have been applied in brain–machine interfaces (BMIs) achieving impressive results [23, 11]. A BMI enables direct interactions with external devices based on brain activities, typically recorded using non-invasive electroencephalographic (EEG) systems. It can provide a communication pathway for severely paralyzed patients or assist in rehabilitation [10]. Besides medical applications, recent developments in wearable devices have pushed BMIs towards consumer-grade products to improve life quality [3], e.g., the Interaxon Muse headband for stress relief [4] or the Emotiv headset for controlling drones [28] and ground vehicles [44]. Safety in BMI systems is paramount [12, 7], because a failure would cause misdiagnoses, user frustration, or even danger while driving a wheelchair or controlling a drone, causing physical and financial damages. Zhang and Wu [43] were the first to show that EEG-based BMIs are vulnerable to adversarial attacks by proposing an unsupervised fast gradient sign method (FGSM) [17]. The work assumes that the acquired signals are sent to a remote compute engine, e.g., a computer, and the attacker can alter

2022 Trustworthy and Socially Responsible Machine Learning (TSRML 2022) co-located with NeurIPS 2022.

the signals during the transmission, between the signal preprocessing step and the classifier. Recent developments in smart edge computing [1, 5] eliminate the need for data transmission, making this attack scenario inapplicable. Novel BMI solutions [21, 40] embed the signal processing and classification directly at the sensor edge. A more practical adversarial example has been identified by [29]. It consists of a square-shaped signal that can be added to EEG trials before the preprocessing step. However, the attack is proposed as a backdoor key, which is improbable if the attacker is not directly involved in the data acquisition or in the training of the classifier. Li et al. [25] have shown an attack scenario in the audio domain by considering the on-board edge processing of a wake-word detection system, where an adversarial audio trace is delivered to the environment causing the system to malfunction. No similar studies can be currently found in the BMI domain.

**Challenges: Designing natural attacks and modeling its propagation.** Unlike in audio applications where the signal can simply propagate over-the-air and is sensed by a microphone, extra modeling is required to evaluate the signal propagation in BMIs based on the physical properties of the biological tissues. In this work, we consider a practical attack scenario where the adversarial perturbations are introduced at the *source* of the data acquisition, as showcased in Appendix A, Figure 5. This can be achieved, for example, via electromagnetic waves delivered to the environment [12] or via current stimulation delivered to the scalp [8, 14]. The perturbations translate into electrical signals propagating over the scalp and are sensed by the electrodes in addition to the EEG signals.

To guarantee the imperceptibility of the attacks, previous works in BMIs create perturbations that are small in amplitude [43, 19, 26], limiting the attack success rate (ASR). Increased perturbation's amplitude yields higher ASR [29], but makes the attack more easily detectable. Moreover, the generated perturbations are square-shaped, which is implausible for biosignals. [18] are the first to observe square-wave artifacts in biosignals' attacks and propose smooth perturbations for electrocardiograms (ECGs). No similar works have been found for designing natural EEG signals.

**This work: Physically-constrained attacks.** To address the above technical challenges, and for analyzing the vulnerability of BMI models in practical scenarios including edge computing, we design a new attack algorithm that generates natural adversarial examples based on the signals' first derivative and model its propagation over the scalp based on a realistic head model by taking into consideration the attack source and the electrical and physical properties of the conducting tissues. We attack the most energy-efficient network that has been embedded on microcontrollers for smart wearable BMIs called EEGNet [23, 37]. It is a resource-friendly convolutional neural network (CNN) and is the SoA in terms of accuracy and energy-efficiency trade-off [6, 27, 40, 37].

We evaluate our methods and show experimental results on BMIs based on the motor imagery (MI) paradigm. By imagining the movement of different body parts, the decoded intention is translated into control signals. It is widely applied in several BMI applications, such as the control of wheelchairss [41], prosthetic armss [13], ground vehicles [44], and in communication [9]. It has been proven to be the most difficult task to be attacked among the most common BMI paradigms [43, 29]. We evaluate our methods by "fooling" the victim model to always predict the target MI class.

**Main contributions.** Our main contributions are:

- We design a new method to generate natural adversarial perturbations that are similar to original EEG signals and imperceptible to the human eye.

- We model a practical scenario where the perturbation is added at the signal acquisition source and model its propagation constrained by the physical properties of the human scalp.

- The first study of adversarial perturbations in BMI to consider the practical scenario of smart edge computing and physical signal propagation. Our attacks consistently achieve a success rate of $> 50\%$ pointing to the significant vulnerability of the SoA embedded EEGNet.

We hope that our work raises awareness for potential risks, also considering the market tendency of smart wearable devices, and motivates the future development of appropriate countermeasures.

## 2   Modeling natural and physically-constrained attacks

We first introduce a new method to eliminate the square wave artifacts to generate natural perturbations. We then present a practical signal propagation model based on the physical properties of the underlying biological tissues. Appendix B summarizes the background knowledge of BMI classification.

## 2.1 Design and assessment of natural attacks

Projected gradient descent (PGD)-designed attacks on EEG tend to form perturbation signals which resemble a square-wave artifact (see Figure 2), an effect that has been observed on ECG data, too [18]. However, EEG signals are of random nature and can be modeled as frequency dependent stationary or non-stationary random processes [20]. To this end, we introduce a new loss term in the PGD optimization such that the perturbation resembles the random nature of EEG signals, which we achieve by promoting signal changes represented in the first order derivative. We estimate the per-channel derivative $\mathbf{V}' = (\mathbf{v}'_0, \mathbf{v}'_1, ..., \mathbf{v}'_{N_{ch}-1}) \in \mathbb{R}^{N_s-1 \times N_{ch}}$ using the sample-wise difference:

$$\mathbf{v}'_c[t] := \mathbf{v}_c[t] - \mathbf{v}_c[t-1] \quad t \in \{1, 2, ..., N_s - 1\}, \, c \in \{0, 1, ..., N_{ch} - 1\} \tag{1}$$

The additive loss term is determined by $l_1(\mathbf{V}) = -\frac{\beta}{\epsilon} \sum_{c=1}^{N_{ch}} ||\mathbf{v}'_c||_1$, where $|| \cdot ||_1$ is the $\ell_1$-norm, $\epsilon$ the maximum perturbation amplitude, and $\beta \geq 0$ a weighting factor. When designing a one-dimensional perturbation, the derivative loss becomes $l_1(\mathbf{v}) = -\frac{\beta}{\epsilon} ||\mathbf{v}'||$.

**Measuring the plausibility of attacks.** None of the previous works have given quantitative measures to assess the physiological plausibility of an EEG adversarial attack. In this work, we propose data-driven measures for quantifying the naturalism of an attack. We compute the cross correlation and the cosine similarity between the attacked signal and the original EEG, and average the values over the $N_{ch}$ channels and over the samples in the dataset.

## 2.2 Spatial propagation model

We consider the realistic use case where the perturbation signal $\mathbf{v} \in \mathbb{R}^{N_s}$ is emitted from one location, e.g., from an adversarial device placed on the left side of the subject or close to the left ear. More specifically, in this study, we assume that the EEG electrode at the position T9 according to the international 10-10 system [36], which is the closest to the left ear, senses the largest perturbation. The signal subsequently propagates over the skin to each electrode, which results in an individual magnitude and delay depending on the distance between the adversarial device and the electrode. This phenomenon can be experimentally observed on measured EEG traces [30, 35]. More formally, we model the sensed perturbation at channel $i$ and time instant $t$ as

$$h_i(\mathbf{v}, \lambda_m, \lambda_d)(t) := m(l_i, \lambda_m) \cdot \mathbf{v}\left(t - d(l_i, \lambda_d)\right), \tag{2}$$

where $m(l_i, \lambda_m)$ and $d(l_i, \lambda_d)$ are the magnitude and the delay respectively, both of which depend on the distance $l_i$ and on characteristic parameters $\lambda_m$ and $\lambda_d$. We define the resulting multi-channel perturbation $\mathbf{V} \in \mathbb{R}^{N_s \times N_{ch}}$, which is added to the multi-channel EEG signal, as

$$\mathbf{V}(\lambda_m, \lambda_d) = H(\mathbf{v}, \lambda_m, \lambda_d) := (h_0(\mathbf{v}, \lambda_m, \lambda_d), h_1(\mathbf{v}, \lambda_m, \lambda_d), ..., h_{N_{ch}-1}(\mathbf{v}, \lambda_m, \lambda_d)). \tag{3}$$

We estimate the distance $l_i$ between the electrode at position T9 and the remaining, attacked positions using the 10-10 system and a head model with a radius of 8.7 cm [2]. We decouple the distance-dependent modeling of the magnitude and delay, explained in the following paragraphs.

**Magnitude.** We assume that the adversarial device injects or induces a current $I$, yielding a potential $V$ measured near T9. The current propagates over the head surface through the skin to each of the remaining attacked electrodes, which can be modeled as a cylindrical resistor with resistance

$$R_i = \frac{l_i}{\sigma A}, \tag{4}$$

where $\sigma$ is the conductivity of the skin which can be in the range of [0.28, 0.87] Siemens/m [39], and $A$ is the area of the skin conductor. The potential at electrode $i$ is $V_i = V - I \cdot R_i$, and hence the magnitude can be described as

$$m(l_i, \lambda_m) = 1 - \frac{V - V_i}{V} = 1 - \frac{I}{V \sigma A} l_i = 1 - \lambda_m l_i, \tag{5}$$

where we further constrain $0 \leq m(l_i, \lambda_m) \leq 1$. The characteristic magnitude parameter $\lambda_m$ represents the complex interplay between input current, voltage, conductivity, and area, covering various attack scenarios. We consider different characteristic magnitude parameters $\lambda_m \in [1, 15]$. A large $\lambda_m$ represents cases with large attenuation and limited propagation, i.e., a limited set of neighboring electrodes sense the perturbation. Conversely, a small $\lambda_m$ covers cases with lower

attenuation where the perturbation can propagate further and infects all electrodes. We consider also an intermediate case where around half of the electrodes are affected by the attack with $\lambda_m = 5$. Appendix C provides examples of the magnitude of the spatial propagation on the head model.

**Delay.** The propagation of a signal on the head surface yields a position-dependent phase angle or delay, as shown by experimental measurements of related studies [32, 33]. The delay stems from a combination of resistive and capacitive components that are encountered during the propagation of the signal, which can be modeled as an RC-circuit with resistance $R$, capacity $C$, and time constant $\tau = R \cdot C$ that relates to the group delay. Specifically, the contacts between the electrodes and the skin are predominantly capacitive whereas the skin itself is both resistive and capacitive [22]. As explained in the previous part, an increasing distance between the attacker and the target electrode yields a larger resistance $R$. As a result, the time constant $\tau$ and the delay increase too.

Here, we model a linear distance-delay relation. We rely on a study by [32], which conducted human skin impedance and phase angle measurements by placing electrodes at an approximate distance of 10 cm and applying voltages with frequencies in the range 2–1000 Hz. When assuming a linear frequency-phase relation in low-frequency region [33], one can derive the group delay to be 2.8 ms when considering a measured angle of $10°$ at 10 Hz. As those measurements were conducted for only one distance, we extrapolate the delay for the remaining distances using a rectified linear model:

$$\lambda_d \cdot (l_i - l_0) > 0 \ ? \ d(l_i, \lambda_d) = \lambda_d \cdot (l_i - l_0) + d_0 : d(l_i, \lambda_d) = 0, \tag{6}$$

where $d_0 = 2.8$ ms is the delay at distance $l_0 = 10$ cm. The delay depends not only on the distance, but also on other parameters such as the electrode-to-skin contact, the humidity of the skin, etc. To this end, we evaluate the propagation of the attack with different characteristic delay parameters $\lambda_d \in [0.1, 0.563]$ s/m. With $\lambda_d = 0.1$ we cover the cases where very little delay happens, while the largest considered $\lambda_d = 0.563$ s/m yields a maximum delay of 0.1 s at the farthest electrode T10, which is in alignment with the observed EEG measurements [30, 35]. Similarly to $\lambda_m$, we showcase also for an intermediate value of $\lambda_d = 0.3$ which corresponds to a delay of 0.053 ms at T10.

## 2.3 Attack Design

We present imperceptible attacks in BMIs that respect domain constraints such as maximum amplitude, spectral distribution, physiological plausibility, and the spatial propagation of the perturbation. To this end, we formulate a general objective function that contains the spatial propagation, the preprocessing step, and the first order derivative loss term:

$$\mathcal{L}_{tot}\left(\mathbf{X}, \mathbf{v}, \lambda_m, \lambda_d\right) = l\left(H_{bp}\left(\mathbf{X} + H(\mathbf{v}, \lambda_m, \lambda_d)\right), y_{rest}\right) - \frac{\beta}{\epsilon}||\mathbf{v}'||_1, \tag{7}$$

where $l(\cdot, \cdot)$ is the negative log-likelihood loss and $\beta$=1e-6 is a scalar that weights the contribution of the derivative loss term. We compute the perturbation using PGD with $G$=10 iterations, where each iteration consists of a gradient-based update of the perturbation and a projection to the $L_\infty$ ball with radius $\epsilon$ defining the maximum perturbation amplitude. The update rate $\alpha$ is initialized with $\epsilon/2$ and linearly decreased with each iteration, reaching a final value of 0.1 mV at iteration 10. The PGD computation is restarted 5 times with different initial perturbations, which are drawn from a uniform distribution within the range $[-\epsilon, +\epsilon]$. See Appendix D for algorithmic details.

**Propagation model.** We distinguish between three use cases of spatial propagation model, where in all cases an instance-specific attack is computed: Case 1) Ignore the propagation model: a multi-channel perturbation $\mathbf{V}$ is computed, which attacks each channel individually, replacing the terms $H(\mathbf{v}, \lambda_m, \lambda_d)$ by $\mathbf{V}$ and $\mathbf{v}'$ by $\mathbf{V}'$ in equation 7. This case is the same as the channel-specific attacks in [43]. It is used as a comparison baseline and for imperceptibility analyses. Case 2) Consider the propagation model: a single-channel perturbation $\mathbf{v}$ is computed and tested with a specific propagation configuration $\lambda_m$ and $\lambda_d$. With this, we simulate the case where the attacker is aware of the physical properties of the head. Case 3) Consider a use-case where the attacker does not know the spatial propagation model and computes the same perturbation $\mathbf{v}$ for all channels. It is similar to the channel-invariant attacks in [29]. However, the previous study does not consider the physical constraints of the signal propagation. Whereas, in our case, we model the real-world scenario by applying the actual propagation model and demonstrate that, without the physical knowledge, the attacker would fail.

## 3 Experiments and results

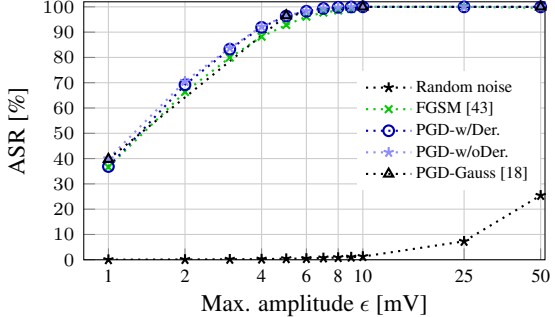

**Figure 1:** Performance of PGD with and without derivative loss term and comparison with random noise and FGSM.

**Table 1:** Plausibility metrics for PGD attack (a) without derivative term, (b) with the derivative loss term, and (c) with a Gaussian kernel [18]. The smaller the cross correlation $\eta$ and the higher the cosine similarity $\gamma$, the more natural the generated attack.

| | $\eta\ [10^{-3}\mathrm{V}^2]$ | | | $\gamma\ [\%]$ | | |
|---|---|---|---|---|---|---|
| $\varepsilon$ [mV] | (a) | (b) | (c) | (a) | (b) | (c) |
| 1 | 3.31 | **1.98** | 3.42 | 99.89 | **99.93** | 99.89 |
| 5 | 16.6 | **7.76** | 17.3 | 97.99 | **99.22** | 97.87 |
| 10 | 32.5 | **12.6** | 34.1 | 93.82 | **97.47** | 93.49 |
| 25 | 74.9 | **27.3** | 79.2 | 79.61 | **90.05** | 78.92 |
| 50 | 125 | **39.0** | 135 | 64.17 | **79.70** | 63.06 |

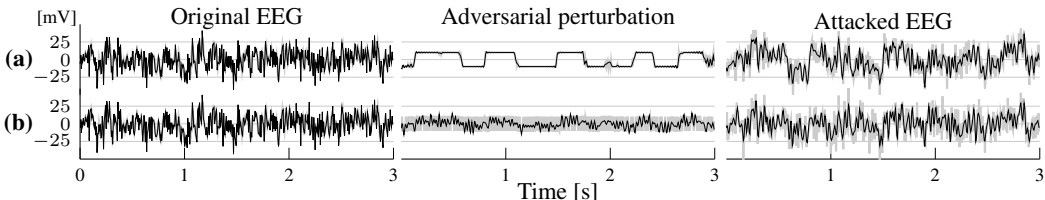

**Figure 2:** A successful attack (a) without and (b) with derivative loss term (PGD, $\epsilon$=10mV). The background traces show the original signal before the preprocessing filter.

We evaluate our methods on the public Physionet EEG Motor Movement/Imagery Dataset [16, 36] tackling the inter-subject variability typical of real-world BMIs. We use a total of 6615 trials with $N_{cl}$=3 balanced classes ("left", "right", and "rest") and obtain the classification accuracy of 74.78% similar to the SoA [40]. Details on the dataset and the training/validation methodology can be found in Appendix E. We then "fool" the classifier to always predict the "rest" class. To determine the ASR, we compute the ratio between the successfully fooled trials and the total number of attacked trials, where we only consider the ones initially correctly classified as "left"/"right".

**Attacks' imperceptibility.** We first analyze the attacks without considering the propagation model (Case 1), depicted in Figure 1. We compare our methods against random noise with amplitude $\epsilon$ as in [43] and FGSM that is the same as in [43] with targeted scenario. For both FGSM and PGD, the ASR increases together with the maximum amplitude $\epsilon$ of the perturbation. They always outperform the random noise, with PGD performing slightly better than FGSM. They reach the maximum ASR of almost 100% with 10mV. The post-attack classification accuracy drops from 74.78% to 48% for a perturbation amplitude of 2 mV and to 33% for 10 mV and higher amplitudes, as reported in Table 2. To be noted that we have a balanced 3-class task, hence, when all samples are classified to one target class, the classification accuracy is 33% (Appendix B).

Figure 2a shows the signals of a successful attack using PGD. The adversarial perturbation has a square-wave form which negatively affects the natural shape of the EEG signal. By adding the proposed derivative term, the square-wave artifacts are significantly reduced (2b), making the perturbation more physiologically plausible. When comparing the power spectral density of the original and attacked signals, the attacked signal designed without derivative presents large components in low frequencies, making it more easily detectable. Whereas the attack with derivative loss better resembles the power spectral density of the original signal (see Appendix F, Figure 7). Moreover, the introduction of the derivative term does not degrade the ASR.

The quantitative measures between the original and the adversarial samples in Table 1 demonstrate

**Table 2:** Pre- and post-attack classification accuracy using PGD with derivative loss term.

| $\varepsilon$ [mV] | ASR [%] | Pre [%] | Post [%] |
|---|---|---|---|
| 2 | 69.18 | 74.78 | 48 |
| 10 | 100 | 74.78 | 33 |

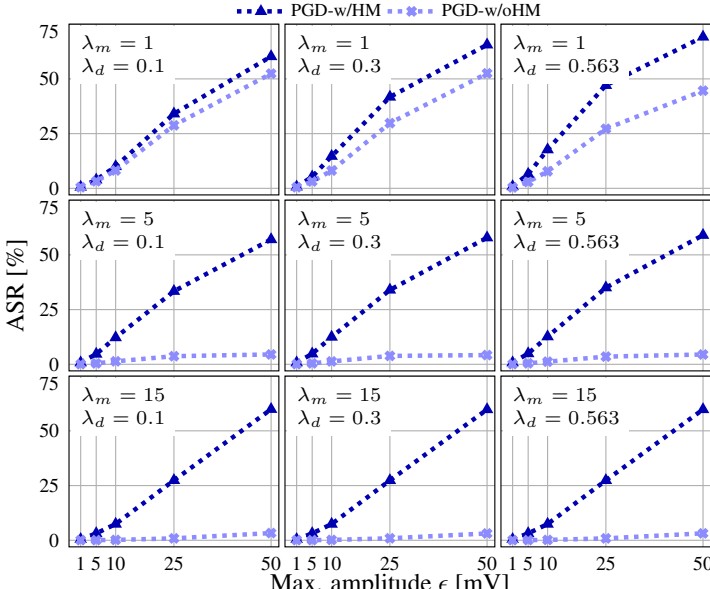

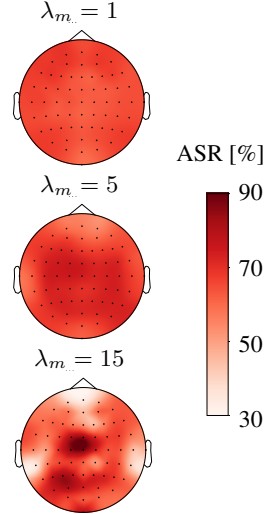

**Figure 3:** ASR of PGD in Case 2), i.e., computed with head model (w/HM), and in Case 3), i.e., computed without head model (w/oHM).

**Figure 4:** ASR with the PGD attack propagating from different EEG channels with fixed $\lambda_d$=0.3 and variable $\lambda_m$.

that our proposed method with derivative term generates adversarial samples that are more similar to the original EEG, allowing them to remain imperceptible even with high $\epsilon$. We reproduce the attacks using a Gaussian kernel as in [18]. After tuning the kernel size and variance of the Gaussian kernel, the method could not improve the plausibility metrics. The inferior performance of the Gaussian kernel could stem from the different nature of the signal: it was originally designed for ECGs which have a pseudo-periodic structure.

**Spatial propagation.** Finally, we introduce the spatial constraints in the signal propagation over the scalp (Case 2). We consider 9 different scenarios by combining 3 realistic attenuation configurations $\lambda_m \in \{1, 5, 15\}$ with 3 delay configurations $\lambda_d \in \{0.1, 0.3, 0.563\}$, which capture the range described in Section 2.2. For evaluating the highest achievable attack efficiency, we test a scenario where the attacker is assumed to know the propagation model: the adversarial perturbation is generated and evaluated on fixed spatial parameters $\lambda_m$ and $\lambda_d$, shown in Figure 3, where the ASR reaches up to 69.2% at 50mV. Figure 8 in Appendix F depicts an example of a successful attack with the highest perturbation amplitude. The introduction of the spatial constraints makes the attack problem harder yielding seldom square distortions. However, the resulting EEG signals still resemble physiological random processes typical of EEGs.

Next, we ablate the spatial constraints during generation and test the resulting perturbations on the 9 above-mentioned scenarios (Case 3). The ASR drops significantly, especially for $\lambda_m$=5 and $\lambda_m$=15 where the attenuation of the perturbation over the scalp is greater. These results support the necessity of physical modeling while designing robust models against adversarial attacks in real-world BMIs. Attacks designed without considering the physical constraints will likely fail in a real-world setup (Case 3). However, if an attacker would model the physical properties of the signal propagation, then the ASR would be significantly increased, yielding a failure in the BMI system (Case 2).

Finally, our spatial propagation models allows us to identify the vulnerability of the individual EEG channels. Figure 4 shows the ASR when initiating an attack from a specific channel (T9, T10, etc.) and propagating it to the rest of the head. In the case with the greatest attenuation ($\lambda_m$=15) we find the maximum ASR at the electrode Cz between the regions of the electrodes C3 and C4, which are the most relevant ones for MI of the left and right hand tasks [31].

## 4 Conclusion

With the incentive of improving security in BMIs, in this work, we demonstrated that PGD-based attacks are feasible and effective if an attacker would consider physical domain constraints. Exper-

imental results reveal potential risks of imperceptible attacks at the signal acquisition source and incentivize the need for future development of defense mechanisms while designing trustworthy and reliable deep learning models not only for remote processing pipelines, but also for the emerging smart wearable BMIs. Our detailed analysis on each EEG channel shows that special attention has to be paid, combined with the findings in neuroscience, to the brain regions that are found responsible for a specific task. In future work, we will investigate physically-constrained attacks in a more realistic scenario where the attacker does not have access to the attacked EEG signals and design subject-independent universal attacks. Moreover, we will evaluate more complicated modelling and introduce uncertainty in the propagation model and the timing of the MI activity.

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

# A    Attack at the source of signal acquisition

Fig. 5 illustrates the new attack scenario where the perturbation is delivered to the human scalp and propagates to the sensing electrodes at the source of the signal acquisition.

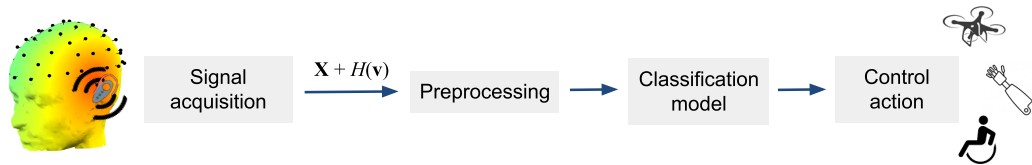

**Figure 5:** Practical adversarial attack scenario in BCIs: a smart device close to the ear emits a perturbation signal which propagates over the head surface to the EEG electrodes.

# B    Classification in BMIs

Here we describe the commonly used approach in BMIs for classification, consisting of a preprocessing step and a classifier. The brain activity is recorded with an EEG device which samples $N_{ch}$ channels at rate $F_s$. We define one trial $j$ as $(\mathbf{X}^{(j)}, y^{(j)})$, where $y^{(j)} \in \{0, 1, ..., N_{cl} - 1\}$ is the true label of $N_{cl}$ MI tasks, and $\mathbf{X}^{(j)} \in \mathbb{R}^{N_s \times N_{ch}}$ the multi-channel recording defined as

$$\mathbf{X}^{(j)} := \left( \mathbf{x}_0^{(j)}, \mathbf{x}_1^{(j)}, ..., \mathbf{x}_{N_{ch}-1}^{(j)} \right), \tag{8}$$

with $\mathbf{x}_i^{(j)} \in \mathbb{R}^{N_s}$ corresponding to the recording of the $j$-th trial and the $i$-th channel containing $N_s$ temporal samples. For simplicity, we denote $\mathbf{X} := \mathbf{X}^{(j)}$ and $y := y^{(j)}$.

The EEG recordings are often preprocessed with a band-pass filter, e.g., using a Fast Fourier Transform (FFT) filter $h_{bp}(\cdot)$, before being fed to a classifier, yielding

$$\mathbf{X}_{bp} = H_{bp}(\mathbf{X}) = (h_{bp}(\mathbf{x}_0), h_{bp}(\mathbf{x}_1), ..., h_{bp}(\mathbf{x}_{N_{ch}-1})). \tag{9}$$

Finally, the preprocessed signal $\mathbf{X}_{bp}$ is classified with a trainable model $f$ and is mapped to $\mathbf{p} := f(\mathbf{X}_{bp})$, where $\mathbf{p} \in \mathbb{R}^{N_{cl}}$ contains the output probabilities, e.g., originating from a softmax activation as final operation in $f$. The model's final prediction $\hat{y}$ is the index with the maximum score in $\mathbf{p}$:

$$\hat{y} = \hat{f}(\mathbf{X}_{bp}) = \underset{y \in \{0, ..., N_{cl}-1\}}{\operatorname{argmax}} f(\mathbf{X}_{bp})[y]. \tag{10}$$

**Classification accuracy.**    The classification accuracy is calculated as the ratio between the number of correctly classified samples and the number of total samples. In a balanced three-class problem, i.e., each class contains the same number of samples, if all samples are classified to one target class, the classification accuracy will be 33%.

# C    Spatial propagation models

Fig. 6 illustrates the magnitude of signal propagation using different propagation parameters $\lambda_m = \{1, 5, 15\}$. A large $\lambda_m$ represents cases with large attenuation and limited propagation (e.g., attack over the air) and a small $\lambda_m$ covers cases with lower attenuation where the perturbation can propagate farther (e.g., a smart glass).

# D    End-to-end algorithm

The end-to-end algorithm based on PGD is illustrated in Algorithm 1.

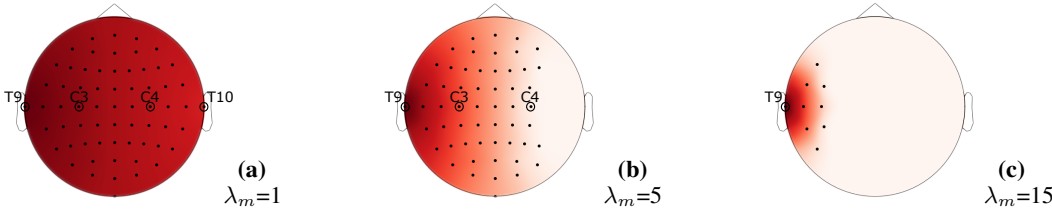

| (a) | (b) | (c) |
|:---:|:---:|:---:|
| $\lambda_m=1$ | $\lambda_m=5$ | $\lambda_m=15$ |

**Figure 6:** Magnitude of the spatial propagation for different $\lambda_m$. The perturbation is emitted from the left side of the head and propagates over the head surface. The leftmost electrode senses the highest magnitude (red), which linearly decreases towards zero (white) with growing propagation distance and $\lambda_m$. The electrodes which sense the perturbations, i.e., magnitude >0, are marked with dots. The electrodes T9, C3, C4, and T10 are labeled for reference.

---

**Algorithm 1:** Generation of imperceptible, physically-constrained PGD attacks.

**input** : $\mathbf{X}$, Original EEG sample; $\lambda_m, \lambda_d$, spatial propagation parameters; $\beta$, weight of derivative loss term; $\epsilon$, maximum perturbation amplitude; $G$, number of PGD iterations

**output** : $\mathbf{X}^*$, adversarial example

1   $\mathbf{v} \leftarrow \mathcal{U}\left(-\epsilon, \epsilon\right) \in \mathbb{R}^{N_s}$;          // Initialisation

2   $\alpha \leftarrow \frac{\epsilon}{2}$;

3   **for** $g \leftarrow 1$ **to** $G$ **do**

4      $\mathbf{V} \leftarrow H(\mathbf{v}, \lambda_m, \lambda_d)$;          // Spatial propagation

5      $\mathbf{p} \leftarrow f(H_{bp}(\mathbf{X} + \mathbf{V}))$;          // Model pass with perturbation

6      $\mathbf{v} \leftarrow \mathbf{v} - \alpha \cdot \text{sign}\left(\nabla_{\mathbf{v}}\left(l(\mathbf{p}, y_{rest}) - \frac{\beta}{\epsilon}||\mathbf{v}'||_1\right)\right)$;          // Update w/derivative

7      $\mathbf{v} \leftarrow \text{clip}_\epsilon(\mathbf{v})$;          // PGD projection

8      $\alpha \leftarrow \frac{0.1 - \frac{\epsilon}{2}}{G} \cdot g + \frac{\epsilon}{2}$;          // Learning rate update

9   **end**

10   $\mathbf{X}^* \leftarrow \mathbf{X} + H(\mathbf{v}, \lambda_m, \lambda_d)$;          // Spatial propagation and add to sample

---

## E   Dataset, training and validation

The Physionet EEG Motor Movement/Imagery dataset contains EEG recordings of 109 subjects [11]. Following the practice by [11], four subjects are discarded due to variability in the number of trials, resulting in the recordings of 105 subjects to be used in our experiments. We use the MI recordings containing tasks of the imagination of left against right fist for 3 s. The EEG trials were recorded with $N_{ch}$=64 channels sampled at $F_s$=160 Hz, yielding $N_s$=3·160=480 samples per trial. Additional baseline runs provide resting-state data, where the subjects did not perform any tasks while having eyes open. Overall, we have a total of 6615 trials with $N_{cl}$=3 balanced classes "left", "right", and "rest."

**Training and validation.** We train and validate the classification models with a 5-fold cross-validation, splitting the dataset into 84 subjects used for training and 21 subjects used for validation to effectively test the model on inter-subject variability. Similar to [40], which achieved SoA performance on this dataset, the baseline model is trained for 100 epochs using Adam with $\beta_1$=0.9, $\beta_2$=0.999, and a batch size of 16. The learning rate is 0.01 and decreased by a factor of 10 at epochs 20 and 50, achieving an average accuracy of 74.78%. An FFT band-pass filter $h_{bp}$ with a customary passband of 0.1–40 Hz [23] is used as preprocessing step in both baseline and attack experiments.

## F   Additional supporting results

**Attacks' imperceptibility.** When comparing the power spectral density of the original and attacked signals, the attacked signal designed without derivative presents large components in low frequencies, making it more easily detectable. Whereas the attack with derivative loss better resembles the power spectral density of the original signal (see Figure 7).

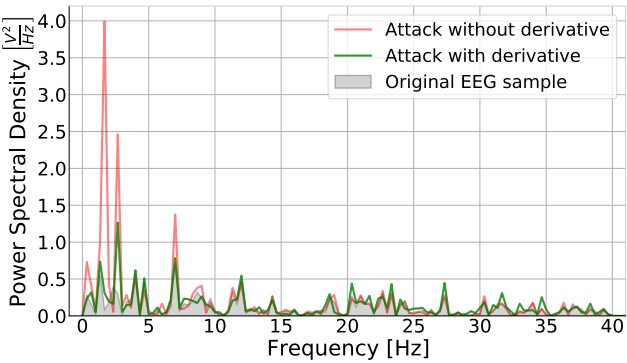

**Figure 7:** Power spectral density comparison of the attack with and without derivative loss term, as well as the original signal shown in Figure 2.

**Spatial propagation.** Figure 8 depicts an example of a successful attack with the highest perturbation amplitude. The introduction of the spatial constraints makes the attack problem harder yielding seldom square distortions. However, the resulting EEG signals still resemble physiological random processes typical of EEGs.

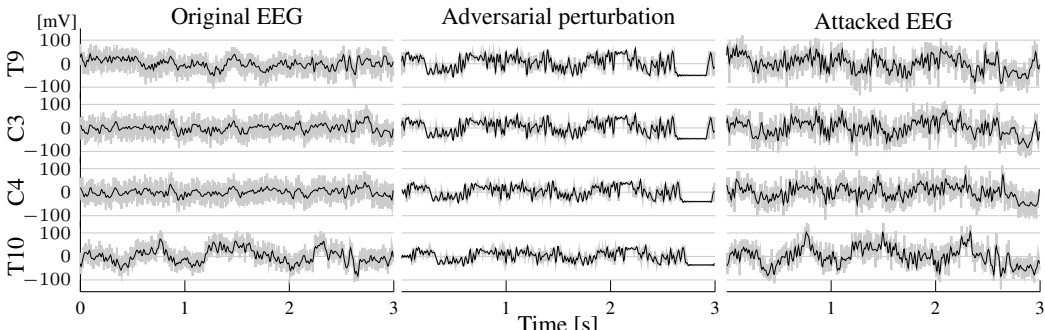

**Figure 8:** A successful attack with derivative loss term and spatial constraints $\lambda_m = 1$ and $\lambda_d = 0.563$ (PGD, $\epsilon$=50mV). The background traces show the original signal before the preprocessing filter.

