# OpenReview forum: "Physically-Constrained Adversarial Attacks on Brain-Machine Interfaces"
_NeurIPS.cc/2022/Workshop/TSRML — TSRML2022_

### Official Review · Reviewer_MbJg · 2022-10-11
**Interesting question, but limited evidence to support its effectiveness**

**Overall Rating:** 5

**Summary:**

The paper studies the physically-constrained adversarial attacks against BMI models. It uses constrain perturbation derivative to make the signal more natural. It further considers a propagation model to model malicious signals that can be injected by a device around the left ear.

**Strengths:**

1. good efforts in designing physically-constrained adversarial attacks against BMI models. It is a practical problem to study

**Weaknesses:**

1. The paper only provides a theoretical analysis of the physically-constrained attacks. However, there is no real-world experiments to demonstrate that the considered physical constraints are useful and effective in practice.
2. Also, the paper did not justify why we should only focus on these constraints. Are there other constraints in the physical attacks?


**Overall Recommendation:**

The paper analyzed some physical constraints in attacking BMI models. However, it did not provide any evidence to support that these physical constraints are practical and useful in the physical world. Therefore, the effectiveness of the proposed approach is questionable, and I am recommending a rejection.

**Review Confidence:**

3: The reviewer is fairly confident that the evaluation is correct

---

### Official Review · Reviewer_Kzx8 · 2022-10-18

**Overall Rating:** 7

**Summary:**

In this paper, the authors studied the adversarial attack on EEG signal data. Specifically, the authors proposed to find a domain-specific attack that can be implemented on wearable devices. The proposed method is evaluated on a real-world EEG dataset as the authors can find physically meaningful adversarial attacks.

**Strengths:**

-Overall, this paper is well-written and easy to follow.
-The motivation of this study is quite valid as there is a growth in monitoring health biometrics with wearable devices.
-I believe the experimental evaluations in this paper are quite sufficient.

**Weaknesses:**

The authors may consider putting the result of BMI classification task in a table (clean accuracy and robust accuracy given different kinds of attacks) in order to better illustrate the effectiveness of the proposed attack.

**Overall Recommendation:**

I believe this paper is addressing a critical problem and should be published in this form.

**Review Confidence:**

4: The reviewer is confident but not absolutely certain that the evaluation is correct

---

### Decision · Program_Chairs · 2022-10-23

**Decision:**

Accept

**Comment:**

DL adversarial robustness for BMIs is a very novel and relevant topic in trustworthy machine learning. The submission provides the first study on this topic and identifies the robustness threat by proposing a strong and imperceptible attack. However, reviewers identified some issues, e.g., the practicality of the physical constraints and whether there exist other constraints. Overall, the submission is accepted, and we strongly recommend the authors address reviewers' comments in the camera-ready version.